# A Qualitative Study on the Use of the Hospital Safety Index and the Formulation of Recommendations for Future Adaptations

**DOI:** 10.3390/ijerph20064985

**Published:** 2023-03-12

**Authors:** Hamdi Lamine, Alessandro Lamberti-Castronuovo, Prinka Singh, Naoufel Chebili, Chekib Zedini, Nebil Achour, Martina Valente, Luca Ragazzoni

**Affiliations:** 1CRIMEDIM—Center for Research and Training in Disaster Medicine, Humanitarian Aid and Global Health, Università Del Piemonte Orientale, 28100 Novara, Italy; 2Department for Sustainable Development and Ecological Transition, Università Del Piemonte Orientale, 13100 Vercelli, Italy; 3Ibn El Jazzar Medical Faculty of Sousse, University of Sousse, Sousse 4002, Tunisia; 4Urgent Medical Aid Service (SAMU 03), Sahloul University Hospital, Sousse 4052, Tunisia; 5School of Allied Health, Faculty of Health, Education, Medicine and Social Care, Anglia Ruskin University, East Road, Cambridge CB1 1PT, UK

**Keywords:** Hospital Safety Index, disaster preparedness, World Health Organization, hospitals

## Abstract

The Hospital Safety Index is a tool developed by the World Health Organization and the Pan American Health Organization in 2008 and updated in 2015. Although it is the most widely used instrument of its kind to assess the level of hospital preparedness, scientific literature on its application in real life is scarce. This study aimed to investigate the use of the Hospital Safety Index to assess disaster preparedness in healthcare facilities. A retrospective, qualitative study employing semi-structured online interviews was conducted to gather the opinions and perspectives of professionals who have experience in applying the Hospital Safety Index. Authors of scientific publications using the Hospital Safety Index were recruited. A semi-structured interview guide was developed. It addressed different phases of data collection with the Hospital Safety Index, the challenges and facilitators of using it, and recommendations for future adaptations. Data were analysed using inductive thematic analysis. Nine participants who were from three countries (Serbia, Sri Lanka, and Indonesia) and had different professional backgrounds (medical doctors, engineers, spatial planners, etc.) participated in this study. A total of 5 themes and 15 subthemes emerged during data analysis. Most of the participants reported their reasons for choosing the Hospital Safety Index as being its comprehensiveness and the fact that it was issued by the World Health Organization. The tool appears to be very specific and allows investigators to spot details in hospitals; however, it is not easy to use, and training is highly encouraged to learn how to navigate the different components of the tool. Governmental support is a crucial facilitator for investigators to be able to enter hospitals and conduct their evaluations. Overall, the tool has a lot of potential, and it should be used to reach a broader audience, such as community members, and assess the preparedness of other facilities that can take part in the response to disasters (hotels, stadiums, schools, etc.). Nevertheless, it still needs more adaptations to be tailored to different contexts and settings.

## 1. Introduction

Over the last few decades, health systems have witnessed the effects of different types of disasters. As a result, disaster preparedness programs have evolved, adopting a more comprehensive, all-hazard approach [1]. Healthcare facilities (HFs) have a central role during disasters, as the number of deaths in a disaster does not depend only on the severity of the event, but also on the ability to effectively respond to it [2]. Although the preparedness of HFs is considered a national security priority and was globally agreed upon to be a target for improvement, as per the Sendai Framework for Action 2015–2030 and the Sustainable Development Goals, the COVID-19 pandemic revealed the inefficiency and lack of preparedness of HFs and sparked a renewed interest in building their capacity for future shocks.

Preparing HFs for disasters has several benefits, such as the protection of lives; the enhancement of effective response and recovery, including continuity of care; the protection of investments; and the reduction in environmental impacts on the health sector [3]. However, preparing HFs for disasters is not an easy task given the complexity of services provided; dependence on water, medical gasses, or waste collection; and the presence of hazardous materials and dangerous equipment [4].

To strengthen the disaster preparedness of HFs, tools and resources have been promoted that support the prioritization of interventions [5]. In the wake of this commitment, 168 United Nations Member States endorsed the Safe Hospital Initiative, committing to build new hospitals in a way that will guarantee continued and safe operation during disasters. Within this initiative, the Hospital Safety Index (HSI) was developed [6]. After the release of its first version in 2008, health authorities worldwide manifested a growing interest in the tool, which became the most widely used instrument of its kind [7]. The extensive use of the HSI led to several adaptations and, ultimately, to the formulation of a second version in 2015 [6]. This latest version of the HSI is used to establish a preliminary diagnosis of the safety of the hospital and its ability to provide services in the event of an emergency or disaster. The tool contains 151 items, each with three levels of security rating: low, medium, and high. It has 4 modules, module 1: ‘Hazards affecting the safety of the hospital and the role of the hospital in emergency and disaster management’, module 2: ‘Structural safety’, module 3: ‘Non-structural safety’, and module 4: ‘Emergency and disaster management’ [6].

Although it is estimated that the HSI has been used to assess the safety of more than 3500 facilities worldwide [6], scientific publications on it are scant. Most of the studies found in the literature refer to a few countries, and they are either reporting the results of HSI assessments [8,9,10,11,12,13,14,15,16], comparing preparedness across cities [17], or adapting the tool to specific contexts [18,19,20]. There is a lack of studies on the methodological implications of using the HSI as a primary data collection tool and a shortage of qualitative studies. Because of the importance that healthcare disaster preparedness has gained recently, and the new awareness of governments, health professionals, and the general population regarding disaster preparedness in the post-COVID-19 era, it is likely that the HSI will be used more frequently in the future.

This study aimed to explore how the HSI has been used to assess disaster preparedness in HFs around the world, understand the challenges and facilitators of its use, and suggest future adaptations for the tool. Specifically, the study aimed to answer the following research questions: How was the HSI used to assess disaster preparedness? What challenges and facilitators did evaluators encounter using the HSI? What future adaptations of the HSI do they recommend?

## 2. Materials and Methods

A qualitative study employing semi-structured online interviews was conducted to collect the opinions and perspectives of professionals using the HSI to assess the disaster preparedness of HFs. This design was chosen because it can properly document the experiences of the users applying the HSI and because of the exploratory nature of the study [21,22]. The study methods have been reported in accordance with the Consolidated Criteria for Reporting Qualitative Research (COrEQ) [23].

The criterion considered for the selection of participants was whether they had published a scientific paper using the HSI. A preliminary review of the literature was performed on Pubmed (the most extensively used database and search engine in the biomedical and healthcare fields), using the keyword ‘Hospital Safety Index’. A list of 19 original research papers was retrieved. All of the authors from those papers were contacted via email and were provided with information on the study objective, methodology, and ethical implications. Upon confirmation, the researchers scheduled interviews with participants. Online interviews were conducted in English.

### 2.1. Data Collection

An interview guide, made of leading and probing questions, was elaborated (see Appendix A). The guide was piloted on a pool of fellow researchers and then adjusted following their feedback. The same guide was used for all interviews, with minor linguistic adaptations when needed. Interviews were conducted between January and April 2022, and each lasted approximately 60 min. After receiving consent from the respondents, an audio recording was made of each interview and manual notes were taken. 

### 2.2. Data Analysis and Reporting

Interviews were transcribed using Sonix software. The transcripts were manually screened and read multiple times for accuracy. Four researchers (HL, MV, ALC, and PS) prepared a list of potential codes, using inductive reasoning, after reading all the transcripts and extracting the most important topics addressed by the participants. The four lists were then compared and a unified codebook was devised to analyse interviews. The unified codebook was used to deductively code each of the transcripts, extracting significant quotes related to the generated codes. To ensure the consistency and reliability of the coding, at least three out of the four researchers were present for the coding of each transcript. A time of approximately three hours was allocated for the analysis of each interview. The analysis was conducted over the course of three weeks.

### 2.3. Ethical Considerations

The study was conducted according to the principles enunciated in the Declaration of Helsinki. All participants were required to provide oral informed consent prior to data collection. Sufficient details about the study aims and processes were provided at the beginning of each interview. The collected data were anonymized, and access to the data was restricted to the co-authors of this paper only. This study was ethically approved by the ethical committee of Habib Thameur Teaching Hospital, Tunisia, under the reference number HTHEC-2021-09.

## 3. Results

More than 40 potential participants from 5 countries (Indonesia, Iran, Mexico, Serbia, and Sri Lanka) were identified; out of these, only 21 contacts were found, and those 21 people were invited by means of email. Nine participants (randomly re-named as A1-9) were accepted for recruitment into this study. A total of 5 themes and 15 subthemes emerged during data analysis. An overview of the demographics of the participants, alongside information about their research, is presented in Table 1.

Participants predominantly used the 2015 version of the HSI. Of these, five did not introduce any modifications to the original version. The distribution of HFs assessed with the HSI is shown in Figure 1.

### 3.1. Use of the HSI

#### 3.1.1. Reasons for Choosing the HSI

Most of the participants (6 out of 9) stated that the reasons for which they had chosen the HSI over other tools were twofold: it was issued by a trustworthy organization (i.e., the WHO) and because of its comprehensiveness. Conversely, other participants reported that the HSI was recommended by their supervisors. Additional reasons for choosing the HSI over other instruments were its use in nearby countries, its availability in the local language, and its relevance to the researchers’ background and objectives.

#### 3.1.2. Purpose of the Assessment

Investigating hospitals’ and public buildings’ disaster preparedness was the most mentioned purpose for using the HSI (5 out of 9). Additionally, respondents mentioned having used the HSI to test its appropriateness in a specific setting (e.g., a specific country or a specific health system), to investigate the possible need for a hospital renovation, or to pursue hospital accreditation.

#### 3.1.3. Before Data Collection 

Almost all participants (8 out of 9) stated that they performed a literature review to gain knowledge about the HSI, and 7 out of 9 participants formed a multidisciplinary team (as per the recommendations of the guide for evaluators) and had peer discussions to analyse the tool in depth. Some participants conducted mock assessments under supervision, while others translated the measure into their local language to facilitate its utilisation.

#### 3.1.4. During Data Collection

The methods of data collection varied between participants. Visual inspection was the most common method of data collection reported by respondents: ‘*We took pictures. We went to the whole area of the building. We were literally going to every space within the building*’ (A3). Seven out of nine participants conducted interviews with hospital staff to gain information about hospital safety. Interviews were mostly conducted in person, though some were conducted through mobile phones or an online platform. One participant adopted focus group discussions, while another one conducted additional interviews with individuals living around the hospital. Some authors used the HSI as a self-administered checklist for the hospital staff.

### 3.2. Strengths of the HSI 

‘HSI is a really good guideline, you can be in a hospital a hundred times, but when you have the guideline with you, you can actually see many invisible things that you didn’t see before’ (A2). Indeed, all participants agreed that being comprehensive and a user-friendly tool were the most important strengths of the HSI. Some participants stated that its flexibility (4 out of 9), ease of adaptability to different contexts (3 out of 9), conciseness, and cost-effectiveness (3 out of 9) were other considerable strengths. Further, 5 participants mentioned the easiness of analysis using the Excel calculator as an additional strength of the HSI assessment. 

### 3.3. Weaknesses of the HSI

Many participants (6 out of 9) highlighted that the HSI is tailored around the US context and might, therefore, be inadequate for non-US settings, given the different health systems and disaster profiles. According to some participants, the HSI is too detailed, and it relies too much on information collected from the staff. Other weaknesses identified by respondents were the non-applicability of the tool to all hazards (such as COVID-19), the inadequate grading system, and the discrepancies between the tool’s modules and the initial hazard assessment, as explained by one participant: ‘*you don’t see in HSI so much connection between the modules, you cannot understand how the characteristics of natural hazards that a building is prone to* (in module 1)*, is related to elements of the other modules* (2, 3 and 4)’ (A1).

### 3.4. Factors Affecting the Use of the HSI

#### 3.4.1. Challenges before Data Collection

Three main challenges before collecting data using the HSI were expressed by the participants. First, the lack of interest from the hospital staff in conducting the assessment. In the participants’ opinion, staff did not perceive the assessment of hospital safety as a priority. Second was the difficulty of obtaining necessary permissions to access the healthcare facility, since public hospitals ‘*are afraid that if they have a low classification, it will alter their reputation*’ (A7). Third was the unavailability of data/documents for preliminary review and the lack of similar research against which to compare results. 

#### 3.4.2. Challenges during Data Collection

According to the participants, most challenges during data collection were related to the quality and accessibility of data. In fact, up-to-date data were either unavailable; inaccessible; considered confidential by hospital staff and, hence, not shared; or not properly archived. This made the task of collecting data even more challenging. At times, the data collection process was regarded as time-consuming because of the amount of data to be collected. 

Some challenges were related to the hospital staff—mainly due to their lack of availability to take part in the assessment and their lack of trust towards the research team. Lastly, some challenges were related to the healthcare facility. Participants mainly complained about restricted access to some parts of the building—a fact that sometimes made the data collection harder or even impossible.

#### 3.4.3. Facilitators

Many participants (7 out of 9) mentioned support via the endorsement of the government health authorities as a facilitator of the conduct of assessments using the HSI: ‘*Once the hospital staff knows that this is under the purview of the ministry as well as the hospital directors, it is not that difficult to get the information*’ (A4). A good example of governmental support was the use of the HSI for hospital accreditation in Indonesia. Raising awareness for the importance of such an assessment amongst the facility’s staff, gaining the trust of the hospital staff, having someone on the team who is a staff member, and the availability of the tool in the local language were mentioned as other facilitators for using the HSI. 

### 3.5. Modifications to the HSI

#### 3.5.1. Modifications Implemented

Many participants implemented changes to the HSI, either by removing or adding questions. For example, two participants reported that they removed non-applicable questions, while another participant added new questions related to the involvement of the community. Some participants used a different technique of data collection. While the HSI is a checklist to be filled in by evaluators, some participants (2 out of 9) used it as a self-administered questionnaire to be filled out by the hospital staff. Lastly, one participant preferred using R programming (i.e., a programming language for statistical computing and graphics) to analyse data instead of the available Excel calculator. 

#### 3.5.2. Modifications Suggested

Three participants stated that ‘*The HSI needs fine-tuning to the local conditions*’. Similarly, other participants suggested adding elements regarding spatial data or staff preparedness. Many participants suggested having two different tools for public and private hospitals because of the differences they have in terms of governance and aims. Furthermore, improvements in the grading system of the HSI were also suggested: ‘*Instead of just three grades, maybe we could have from zero to five or zero to ten*’ (A2). Lastly, the use of new technologies in the assessment methods, such as a web-based tool or a mobile application, was suggested. 

### 3.6. Recommendations for the Future

#### 3.6.1. Recommendations for Researchers

Participants’ suggestions addressed three elements: the team, the facility, and the data collection process. Regarding the team, participants suggested that researchers prepare their teams, either by creating a multidisciplinary team or by training data collectors: ‘*Everyone who collects the data should be trained and have the same perception on how to collect the data*’ (A5). Regarding the facility, participants recommended notifying the staff and asking for documentation beforehand, having good communication with the hospital and, having written consent from the hospital director. Lastly, considering the data collection process, participants suggested allowing plenty of time and using new technologies (such as smartphone apps) or, alternatively, combining interviews, document analysis, and field visits. Moreover, some participants suggested comparing results with other studies that used HSI. 

#### 3.6.2. Recommendations for Hospital Staff

Participants suggested using the HSI as a guide to enhance the safety of the facility, rather than using the tool merely for assessment: ‘*Recommendation must be implemented. Otherwise, there is no point*’ (A4). Moreover, some participants (3 out of 9) recommended conducting the HSI assessment periodically and organizing training sessions for the staff regarding the use of the tool.

### 3.7. Primary Health Care (PHC) Facilities 

Elements concerning the use of HSI in PHC facilities emerged during some interviews. These elements touched on/revolved around the barriers and facilitators of the use of HSI in such facilities. Among the most relevant barriers, the lack of interest in studying PHC facilities was frequently mentioned, as well as the PHC staff’s lack of knowledge about many elements of the HSI. Additionally, some items in the HSI were not relevant to the PHC facility; thus, some participants decided to skip them. On the other hand, as a facilitator, the use of a medium/small version of the HSI was an option for adapting the tool to the PHC context.

## 4. Discussion

This study explored how the HSI has been used to assess disaster preparedness in HFs in different settings and countries, examined the challenges and facilitators of the use of the HSI, and formulated suggestions for future adaptations of the HSI. This work examined the methodological implications of using the HSI and provided practical recommendations to enhance its use. The main findings of this study represent the perspectives of nine participants using different versions of the HSI in different types of facilities.

The HSI proved to be the first choice for assessing hospital disaster preparedness, complementing the findings of Heidaranlu et al. [24]. This is primarily because it is endorsed by the WHO, which strengthens the tool’s accountability and validity for measuring disaster preparedness. In fact, since its creation, the HSI has gained much interest worldwide. It has been used as an assessment tool in more than 37 countries and as an accreditation tool in countries such as Indonesia [9,16,25,26]. Because the tool takes a thorough and comprehensive approach to hospital preparedness, completing an assessment requires careful study and significant time investment. Nevertheless, the scarcity of alternatives results in the HSI rarely being questioned [9,16,25].

Among the most prominent obstacles is the difficulty for researchers to receive approval for conducting assessments of disaster preparedness in hospitals. This is not related to the HSI itself, but rather, it results from the scarce awareness of the importance of disaster preparedness among hospital staff and management boards. It is interesting to note that, in countries where the HSI is used to obtain hospital accreditation, this type of assessment is easier. This indicates that a national hospital accreditation system that considers hospital safety could facilitate the use of the HSI and, thus, increase the chances of hospitals measuring their preparedness level and possibly acting upon it [16].

Hospitals are known to be complex systems that play many roles, such as providing health care, laboratory analysis services, hotel services (catering, laundry cleaning, etc.) and maintaining their buildings (offices, treatment rooms, and warehouses). Some of the study’s participants noted this complexity as a challenge. Indeed, it has been demonstrated that the complexity of hospitals in the United Kingdom has adversely affected their ability to respond to emergencies, such as COVID-19, and has made it more challenging to gauge their level of preparedness [18,25,27,28,29]. The WHO’s guide for evaluators covered some of these concerns, but it did not offer any concrete solutions for how to deal with them [6].

It is noteworthy that all interviewees involved in the current study used the HSI as a tool to meet specific security standards or for accreditation purposes rather than to monitor HF preparedness levels and safety. It is, thus, necessary to increase awareness of the importance of monitoring preparedness via the HSI as well as the benefits that are derived from this among healthcare workers, health managers, and public health experts. This can be done by educating healthcare professionals on the basic principles of disaster medicine and disaster management, promoting the use of the HSI via workshops and webinars, and advocating for institutional pressure on hospital safety, considering the projected increase of disasters in the future [30,31,32].

Overall, the need for effective and evidence-based training of healthcare personnel at all levels, including the development of standards and guidelines for training in multidisciplinary health responses to major events, has been identified as a priority by the disaster response community. Despite training and education being considered integral, findings demonstrated that they were not yet standardized [33,34,35]. 

On a positive note, the WHO, and its regional offices, particularly the East Mediterranean and Europe Regional Offices, collaborated to train local assessors on how to use the HSI. Numerous assessors have emerged because of these activities, many of whom have developed strategic plans to carry out national assessments, thus indicating significant interest in the tool’s adoption by nations and regions. In addition, the Disaster Risk Reduction (DRR) community is increasingly considering hospital preparedness by developing new tools and frameworks, such as the CADRI tool. By doing so, this community is bringing a new perspective of hospital sustainability into hospital preparedness for disasters, and it is being given more and more consideration [36,37,38,39].

Another important finding concerns the lack of space for cultural considerations in the HSI. Findings highlighted how some participants had to adapt the tool to their cultural context and country-specific disaster profile to perform a meaningful assessment. This is relevant considering the different types of health systems and facilities worldwide, as well as different disaster profiles. Moreover, of relevance is the fact that hospitals take on different forms in different societies, constituting a space where core cultural values come into view [40]. It follows that it is very important to conceive an instrument for the assessment of hospital preparedness that considers cultural differences in healthcare without compromising the scientific validity of the assessment itself [41]. 

Following a bottom-up approach in assessing disaster preparedness, some respondents emphasized the importance of expanding the spectrum of responses to collect data not only by consulting medical doctors, nurses, and hospital managers, but also by consulting other workers, such as cleaning staff or maintainers. Other respondents consulted nearby communities to retrieve information. These considerations are key for proper Health Emergency and Disaster Risk Management [3]. Overall, consulting people with local understanding and experiential knowledge has the potential to increase the validity of disaster preparedness assessments [42]. 

The scoring system of the HSI has been criticized because of its narrow grading scale, using only three grades (‘low’, ‘medium’, and ‘high’) [6] A wider range of scores (from 1 to 5 or from 1 to 10) might be more effective and easier to implement. 

Additionally, taking into consideration that HFs are contributing to climate change, and thus, to a possible future health emergency, it is the authors’ belief that the sustainability of HFs needs to be factored in as well—whether within the HSI or any assessment of HFs’ preparedness, security, or resilience [43].

In addition to hospitals, the HSI has been used in PHC facilities. Those facilities are inherently different, which is why a modified version of the HSI with fewer elements exists and is documented in the literature. However, it is not well known worldwide and has only recently been used more frequently. Despite the recognized importance of strengthening disaster preparedness at all levels of the health system, literature supporting primary healthcare disaster preparedness is scant and lacks rigour [39,44]. There is no internationally recognized and validated method for evaluating the safety of PHC centres in case of disasters. Even though the HSI covers many important factors, some PHC-specific elements of disaster preparedness may not be captured by the HSI tool (e.g., PHC team composition, continuity of care for patients with chronic diseases, and the assessment of vulnerable categories of patients). Moreover, studies assessing the relevance of the HSI tool for the PHC setting are lacking [20,39]. An operational framework conceptualising specific characteristics of primary care disaster preparedness, categorised according to the WHO health system building blocks [44], was recently published by members of this research team [45]. This can be considered for future adaptations of the HSI, as well as for the creation of primary care-specific tools aimed at strengthening all-hazard preparedness. 

### 4.1. Recommendations

Several recommendations emerged from this study that can be categorised as recommendations for hospitals’ decision-makers, recommendations for researchers, and recommendations for HSI reviewers from the WHO.

#### 4.1.1. Recommendations for Hospitals’ Decision-Makers

Awareness of the importance of assessing hospitals’ disaster preparedness should be raised in hospitals, and decision-makers should promote the use of the HSI to facilitate the identification of weaknesses and inform the allocation of resources. Training, workshops, and awareness-raising campaigns for the importance of performing disaster-preparedness assessments with the HSI should be carried out in hospitals, and incentives should be provided to hospitals that routinely perform disaster-preparedness assessments. This could ultimately facilitate receiving approval for assessments conducted by academic researchers, WHO investigators, or internal hospital boards.The use of the HSI as a guide for the routine monitoring of disaster preparedness in HFs should be promoted. This can be done by establishing trained teams in hospitals that carry out the evaluation and using this as an opportunity to improve hospital preparedness. Overall, the HSI should become a tool for enhancing proactive preparedness strategies rather than an instrument used merely for preparedness evaluations. This could ultimately favour the production of up-to-date reports on the preparedness of HFs and inform policymakers and hospitals’ decision-makers of intervention strategies aimed at strengthening hospital preparedness.

#### 4.1.2. Recommendations for Researchers

More research should be conducted on how to overcome the current limitations of the HSI, including limitations of the scoring system and cultural barriers encountered when using the tool in non-US settings. Those performing disaster preparedness assessments with the HSI are encouraged to highlight, in their publications or reports, the operational challenges that were encountered while using the tool. In turn, HSI reviewers from the WHO are encouraged to take such recommendations into account to improve the tool and facilitate its utilisation. This could ultimately inform HSI reviewers of the most important issues to address when elaborating future versions of the HSI.Research should highlight differences in the preparedness of hospitals, smaller facilities, and primary care centres. Given the variety of different health systems’ infrastructures, it is important to diversify disaster-preparedness assessments by considering peculiar elements of different levels of care. The HSI can be used as a guiding framework for the development of secondary tools that can be used to assess the preparedness of other components of the health system. This could ultimately stimulate the development of secondary tools that allow investigators to assess the preparedness of different health system components.

#### 4.1.3. Recommendations for HSI Reviewers

Future versions of the HSI should take technology into account to ease data collection. For example, a mobile phone application could facilitate data collection and calculations, increasing the chances of the HSI being routinely used. In addition, an online platform monitoring the disaster preparedness of hospitals worldwide can have added value in promoting cross-country comparisons. This could be an incentive for performing assessments with the HSI, even in low-resource settings or by internal hospital staff having limited resources (time, personnel) to deploy for such an assessment. An online platform reporting the results of hospitals’ assessments worldwide could be a key resource for monitoring global improvements in disaster preparedness and sharing best practices.Given the increased attention on hospital sustainability and climate change mitigation worldwide, it is recommended that elements of sustainability are incorporated into the HSI. Considering that the health sector is a major contributor to climate change, integrating disaster preparedness and sustainability principles might lead to several co-benefits. This could ultimately enrich disaster preparedness assessments, rendering them more relevant for environmental scientists and contributing to decreasing hospitals’ environmental impacts.The HSI could be used beyond the health sector to reach a broader audience (such as community members) and to assess preparedness not only for HFs but also facilities that can take part in the response to disasters, such as hotels, stadiums, or schools. When disasters strike, hotels or stadiums can, indeed, be chosen as urban disaster shelters, as they may offer refuge to the affected communities. City buildings, such as hotels or schools, can also be used to store medical equipment and materials when disasters hit hospitals’ infrastructures. It might, therefore, be useful to include in the HSI an evaluation of the preparedness of secondary buildings for a comprehensive assessment of hospital disaster preparedness in a given area.

### 4.2. Strengths and Limitations

The main limitation is the small sample size, as only nine individuals accepted the invitation to take part in this study. Furthermore, the interviewees came from a small number of countries; thus, results cannot be considered generalisable to the world population. It must be said, however, that the duration and depth of the interviews allowed us to reach data saturation and comprehensively answer the research question. The extensive expertise of the research team in hospital disaster preparedness, previous experience using the HIS, and the scientifically sound methodology based on the COrEQ guidelines are strengths that characterise this study. This is the first qualitative study exploring the methodological implications of using one of the most important tools for assessing hospital disaster preparedness and, therefore, has the potential to provide useful insights for future adaptations and modifications of the tool.

## 5. Conclusions

This study demonstrated that the use of the HSI is broad and diverse, depending on the context, the final goal, and the methodologies used. Since it has been 7 years since its last update—and for it to be used in an optimal way—the HSI still needs more adaptations to be tailored to different contexts and settings. Some of the adjustments could be to incorporate new technologies that facilitate data collection and handling; to re-evaluate the grading system to make it broader (from 1 to 5 or 1 to 10); and to make the tool flexible enough to be updated autonomously in accordance with the context and the facility being assessed, by giving the assessor the freedom to remove elements that are unnecessary or to construct their own pool of elements. The HSI has shown its effectiveness in assessing HFs’ preparedness in many settings around the world, yet the users of this tool are still facing challenges in using the tool to its full potential. The authors encourage future revisions of the HSI to take the recommendations elaborated on in this study into account.

## Figures and Tables

**Figure 1 ijerph-20-04985-f001:**
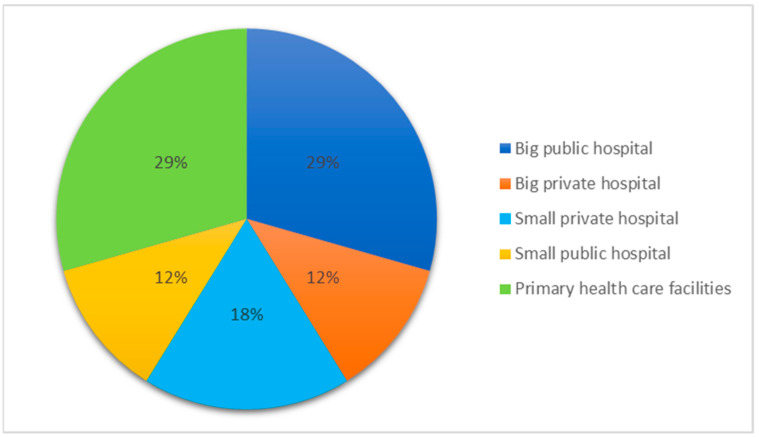
Distribution of health facilities assessed by the participants included in the study.

**Table 1 ijerph-20-04985-t001:** Overview of the demographics of the participants.

Author	Country	Professional Background	Years of Experience Using HSI	Version of HSI	
				Year	Modified
A1	Serbia	Architecture	5–10	2015	No
A2	Sri Lanka	Civil engineering	2–5	2015	No
A3	Serbia	Architecture and urban planning	5–10	2015	No
A4	Sri Lanka	Civil engineering	5–10	2015	Yes
A5	Indonesia	Public health	2–5	2015	No
A6	Indonesia	Occupational health and safety	5–10	2008	Yes
				2015	Yes
A7	Indonesia	Public health	2–5	2008	No
				2015	No
A8	Indonesia	Occupational health and safety	2–5	2015	Yes
A9	Serbia	Medical doctor	2–5	2015	Yes

## Data Availability

The data presented in this study are available on request from the corresponding author. The data are not publicly available due to participant’s refusal.

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
