# Peer review of "A Qualitative Study on the Use of the Hospital Safety Index and the Formulation of Recommendations for Future Adaptations"

_ijerph, 2023, doi:10.3390/ijerph20064985_

Round 1

Reviewer 1 Report

Using semi-structured interviews, this paper discusses past experiences in using hospital safety index to assess health facilities' disaster preparedness. Based on these experiences, recommendations are then made for the further use of the tool. The authors should consider the following comments.

1. In Materials and Methods, the authors should describe and justify the search criteria and keywords used to find the 19 publications. Consider to include the profile of these publications. The use of Pubmed to search publications should also be justified.

2. In 2.1, I suggest that the key questions or the summary of the interview questions are provided so that the research can be replicated and to increase clarity.

3. in Results section, 9 participants participated in the interviews. Please clarify the number of prospective participants contacted and explain why 9 participants participated at the end.

4. Line 129, HIS should have been HSI.

5. Check the last sentence in section 3.1.1. The last several words should be deleted (starting from 'were').

6. It seems that the recommendations are made for researchers and hospitals' decision makers. The recommendations may need to be expanded to reflect the richness of the findings. The recommendations for specific stakeholders should be made clearer, i.e., what can researchers do next and what can decision makers do next and other relevant stakeholders, if any.

7. The conclusion may need to be expanded. It should address the research aim set in the introduction.

8. Research limitations should also be added in the discussion or the conclusion section.

Reviewer 2 Report

Abstract:

Overall, while the abstract provides a general overview of the study, it could have been more informative and specific about the study's methodology, findings, and recommendations. Please do not use abbreviation in abstract.

Introduction

The text provides a good introduction to the topic and effectively sets the stage for the study that is being proposed. One suggestion for improvement would be to provide more detail on the specific research questions that the proposed study aims to answer. This would help the reader to understand more clearly what the study aims to achieve and how it will contribute to the existing literature on disaster preparedness in healthcare facilities.

Methods

Authors provides a clear description of the methods used in the study, including the selection of participants, data collection, and analysis procedures. The use of semi-structured online interviews is appropriate for exploring the experiences of professionals using the HSI to assess disaster preparedness of HFs, and the study design is appropriate for the exploratory nature of the research. The authors have reported the methods in accordance with the COrEQ, which is a good practice.

It is commendable that the researchers contacted all the authors of the 19 papers identified in the preliminary literature review and provided information on the study objective, methodology, and ethical implications. However, it is unclear how many participants were recruited and how they were selected from the contacted authors. Additionally, this section could have provided more information on the characteristics of the participants, such as their professional backgrounds and geographical locations.

The description of the data analysis process is clear and detailed, and the use of a codebook and deductive coding is appropriate for organizing and analyzing the data. The allocation of three hours for the analysis of each interview seems reasonable, but it would have been useful to know how the researchers ensured the consistency and reliability of their coding.

Results

Well written but why authors post recommendations in this section? This should be separated.

Discussion:

In this section authors explored the use of the Hospital Safety Index (HSI) to assess disaster preparedness in healthcare facilities (HFs) in different settings and countries. It also examined the challenges and facilitators of using the HSI and suggested future adaptations. The study found that the HSI is a useful tool for assessing hospital disaster preparedness, but there are obstacles to its use, such as a lack of awareness among hospital staff and management of the importance of disaster preparedness. The complexity of hospitals also presents a challenge, and the study recommends increasing awareness and education of healthcare workers and promoting the use of HSI. The study also identified a lack of cultural considerations in the HSI and suggests developing an instrument for assessing hospital preparedness that considers cultural differences in healthcare. The study recommends expanding the spectrum of responses to include input from a range of stakeholders, including local communities. The study also suggests expanding the scoring system of the HSI to make it more effective and easier to implement. Finally, the study highlights the need to consider the contribution of HFs to climate change and its potential impact on future health emergencies.

Here are some suggestions on how to improve the paper

1.      Provide more context: The paper could benefit from more background information on the Hospital Safety Index and its purpose in supporting the "Safe Hospitals" initiative by WHO. This would help readers who are unfamiliar with the HSI to better understand the significance of the study and its potential implications.

2.      The paper suggests that the HSI could be used beyond the health sector to assess preparedness in an interdisciplinary way, including facilities like hotels, stadiums, and schools. However, this idea could be explored further with examples of how the HSI could be adapted and applied in these settings.

3.      While the paper identifies several potential adaptations to the HSI, it could benefit from more clarity on how these recommendations would be implemented and what their impact might be. For example, how could new technologies be incorporated as assessment techniques, and what benefits would this bring?

4.      The paper briefly mentions some limitations of the study, but could benefit from a more detailed discussion of these. For example, were there any factors that may have influenced the results or the recommendations made?

5.      The paper ends somewhat abruptly without a conclusion, which could leave readers wondering about the significance of the study and its implications. A conclusion could summarize the key findings and recommendations and provide some final thoughts on the potential impact of the study.

Some sugessted reference:

https://doi.org/10.3390/su14031488

Overall, the paper is well-written and provides insightful comments regarding the potential of the HSI tool. It is a good suggestion to expand the use of the HSI beyond the health sector to reach a broader audience and assess preparedness in an interdisciplinary way.

The recommendation to incorporate new technologies as assessment techniques is particularly relevant given the rapid advancement of technology. The idea of re-evaluating the three-level grade system and making the tool flexible enough to be updated autonomously in accordance with the context and the facility to be assessed are also important considerations.

Future studies could benefit from a larger sample size and a more diverse range of participants to provide a more comprehensive understanding of the challenges and facilitators of using the HSI in disaster preparedness assessments.

In summary, the paper provides valuable insights into the potential of the HSI tool and offers practical suggestions for its improvement.

Round 2

Reviewer 1 Report

Thank you for responding to the comments. The revision is detailed. I particularly like the way you expanded the discussion section by the inclusion of recommendations to various stakeholders.